# Metabolomic Profiling of *Citrus unshiu* during Different Stages of Fruit Development

**DOI:** 10.3390/plants11070967

**Published:** 2022-04-01

**Authors:** Sang Suk Kim, Hyun-Jin Kim, Kyung Jin Park, Seok Beom Kang, YoSup Park, Seong-Gab Han, Misun Kim, Yeong Hun Song, Dong-Shin Kim

**Affiliations:** 1Citrus Research Institute, National Institute of Horticultural & Herbal Science, Rural Development Administration, Seogwipo 63607, Korea; sskim0626@korea.kr (S.S.K.); pkj5690@korea.kr (K.J.P.); hortkang@korea.kr (S.B.K.); yspark1219@korea.kr (Y.P.); skhan@korea.kr (S.-G.H.); mkim2019@korea.kr (M.K.); 2Division of Applied Life Sciences (BK21 Four), Gyeongsang National University, Jinju 52828, Korea; yh0126@hanmail.net; 3Department of Agricultural Chemistry, Institute of Agriculture and Life Science (IALS), Gyeongsang National University, Jinju 52828, Korea; hyunjkim@gnu.ac.kr; 4Institute of Animal Medicine, Gyeongsang National University, Jinju 52828, Korea

**Keywords:** *Citrus unshiu*, fruit development, metabolomics, GC-MS, UPLC-Q-TOF MS

## Abstract

Citrus fruits undergo significant metabolic profile changes during their development process. However, limited information is available on the changes in the metabolites of *Citrus unshiu* during fruit development. Here, we analyzed the total phenolic content (TPC), total carotenoid content (TCC), antioxidant activity, and metabolite profiles in *C. unshiu* fruit flesh during different stages of fruit development and evaluated their correlations. The TPC and antioxidant activity significantly decreased during fruit development, whereas the TCC increased. The metabolite profiles, including sugars, acidic compounds, amino acids, flavonoids, limonoids, carotenoids, and volatile compounds (mono- and sesquiterpenes), in *C. unshiu* fruit flesh also changed significantly, and a citrus metabolomic pathway related to fruit development was proposed. Based on the data, *C. unshiu* fruit development was classified into three groups: Group 1 (Aug. 1), Group 2 (Aug. 31 and Sep. 14), and Group 3 (Oct. 15 and Nov. 16). Although citrus peel was not analyzed and the sensory and functional qualities during fruit development were not investigated, the results of this study will help in our understanding of the changes in chemical profile during citrus fruit development. This can provide vital information for various applications in the *C. unshiu* industry.

## 1. Introduction

Citrus fruits are consumed globally and are one of the most important non-climacteric fruit crops in the fruit industry. This is because of their unique favorable taste and a large number of nutritional benefits [1,2]. Many randomized animal and clinical studies have suggested that citrus phytochemicals, including phenolics, flavonoids, limonoids, carotenoids, and volatile terpenes, are positively associated with a reduction in the risk of various health problems, such as inflammation, cancers, and cardiovascular diseases [3,4]. However, these phytochemical profiles are dependent on various factors such as citrus variety, growing environment, and fruit development [5,6,7].

Among these factors, fruit development involves significant changes in primary and secondary metabolites and can be divided into three stages: (I) fruit set and cell division, (II) growth via cell expansion, and (III) maturation [8,9]. Fruit set can be defined as the transition of a quiescent ovary to a rapidly growing young fruit. During fruit growth, juice sac cell enlargement is mostly driven by vacuole expansion, and fruit maturation begins when growth stops. During the fruit development stage of citrus, its qualities, including appearance, nutrition, taste, and biological activity, are significantly changed by changes in metabolite profiles such as sugars, acids, amino acids, flavonoids, and terpenes.

In recent years, transcriptomic, proteomic, and metabolomic technologies have been used to demonstrate the regulation of fruit metabolism during citrus fruit development [5,8,10]. Among these omic technologies, metabolomics is a useful tool for monitoring and comparing changes in phytochemical profiles during fruit development. Phenolic compounds, carotenoids, and primary metabolites of various citrus varieties, including pummelo, Chandler lemon, Akragas, sweet orange, Persian lime, Kinnow, and Guang Chen Pi have been analyzed at different fruit development stages, and the metabolic pathway associated with citrus fruit development has been proposed [8,10,11,12,13].

Among these varieties, *C. unshiu*, commonly known as Satsuma mandarin, is an important easy-peeling mandarin and is widely cultivated throughout the main citrus production zones, including East Asia, Argentina, Australia, Peru, Spain, South Africa, and Uruguay [14]. In particular, *C. unshiu*—which is bud burst in middle of Mar., blooms in May, and is harvested after the middle of Nov. in East Asia [15]—is mostly consumed as a mature fruit, but the immature fruit is also consumed in various products, such as vinegar, jelly, and jams having strong antioxidant activity [16,17,18]. Although previous studies of *C. unshiu* have shown the changes in the phytochemical profiles of *C. unshiu* according to growing regions [19], post-harvest processes [20], and the comparison of the profiles with other citrus varieties [21], the changes in the phytochemical profile of *C. unshiu* during fruit development have not yet been analyzed.

Therefore, in this study, the primary and secondary metabolites of *C. unshiu* fruits harvested at five different developmental stages were analyzed using ultra-performance liquid chromatography–quadrupole time-of-flight mass spectrometry (UPLC-Q-TOF MS), gas chromatography–mass spectrometry (GC-MS), and high-performance liquid chromatography (HPLC) to provide chemical information for various applications in the *C. unshiu* industry. Furthermore, based on the metabolites, a metabolic pathway associated with citrus fruit development was proposed, and the correlation between these metabolite profiles and antioxidant activities was analyzed.

## 2. Results and Discussion

### 2.1. Appearance Change of C. unshiu during Fruit Development

Weight, size, and color of *C. unshiu* fruit were analyzed according to fruit development (Figure 1 and Appendix A). The fruit weight of *C. unshiu* increased from 31.37 ± 3.79 g (Aug. 1) to 111.45 ± 11.52 g (Nov. 16), while the width and length were increased from 3.47 ± 1.92 and 4.16 ± 1.68 cm (Aug. 1) to 6.44 ± 2.71 and 5.20 ± 2.24 cm (Nov. 16), respectively. The color of *C. unshiu* changed from dark green (Aug. 1) to partially orange (Oct. 15) to full orange (Nov. 16). In addition, the citrus color index (CCI), a standard parameter used to determine the color of citrus [21], showed negative values (−15.03 to −16.66) from Aug. 1 (−15.03) to Sep. 14 (−16.66), whereas it showed positive values on Oct. 15 (0.88) and on Nov. 16 (11.59). In general, matured mandarin has a CCI value of 5 or higher [22]. Based on the change in appearance, the fruit development stage of *C. unshiu* used in this study was divided into two stages: a size increase stage (Aug. 1–Sep. 14) and a color change stage (Oct. 15–Nov. 16). In general, fruit development occurs in three stages (cell division, cell expansion, and fruit maturation) [6]. The first three *C. unshiu* fruit (Aug. 1, Aug. 31, and Sep. 14) corresponded to cell expansion (the second stage), and the last two fruit (Oct. 15 and Nov. 16) corresponded to fruit maturation (the third stage).

### 2.2. Total Phenolic Content, Total Carotenoid Content, and Antioxidant Activities

With the advancement in fruit development of *C. unshiu*, the total phenolic content (TPC) and antioxidant activity decreased, whereas the total carotenoid content (TCC) increased (Figure 2). The TPC of *C. unshiu* fruit decreased gradually from 7.27 mg gallic acid equivalents (GAE)/g dry sample (Aug. 1) to 2.23 mg GAE/g dry sample (Nov. 16). By contrast, the TCC of *C. unshiu* fruit showed the opposite result. The highest content was observed on Nov. 16 (30.09 mg β-carotene equivalent [βCE]/100 g), and the content was 4.3 times higher than that on Aug. 1 (6.93 mg βCE/100 g fresh sample). The decrease in TPC during citrus fruit development could be correlated with the increased activity of polyphenol oxidase (PPO), owing to an increase in fruit pH [23] and a decrease in the activity of phenylalanine ammonia lyase, a key enzyme of the phenylpropanoid pathway for the biosynthesis of phenolic compounds during citrus fruit development [24]. By contrast, an increase in TCC might a result of an increase in the expression of carotenoid synthesis genes such as CitPSY, CitPDS, CitZDS, CitLCYb, CitHYb, and CitZE, which are involved in carotenoid accumulation [25,26].

The antioxidant activity of *C. unshiu* fruit during development was measured using DPPH and ABTS radicals (Figure 2C,D). In the DPPH radical assay, the inhibitory concentration 50% (IC_50_) value of the *C. unshiu* flesh gradually increased during fruit development. The highest value was observed in harvested fruit on Nov. 16 (8192.77 mg/mL), and the value was four times higher than that on Aug. 1 (2047.27 mg/mL) (Figure 2C). Additionally, in the ABTS radical assay, the IC_50_ value of *C. unshiu* fruit sharply increased from Aug. 1 (556.07 mg/mL) to Aug. 31 (1001.12 mg/mL) and then slightly increased on Nov. 16 (1181.56 mg/mL) (Figure 2C). The decrease in antioxidant activity during fruit development was positively correlated with a decrease in phenolic compounds. This result indicated that immature fruits are suitable for the production of citrus products having high antioxidant activity. Similar results have been observed during fruit development stages in various citrus fruits, including Pomelo [24], sweet orange [27], Chinese native cultivars [28], and Navel orange [29]. The decrease in phenolic compounds according to the fruit development led to a decrease in their antioxidant activities.

### 2.3. Metabolomic Analysis

The results of metabolite analysis obtained by GC-MS, UPLC-Q-TOF MS, and HPLC (Appendix A) were compared using multivariate statistical analysis, and the discrimination of *C. unshiu* samples was visualized using a PLS-DA score plot (Figure 3). The goodness-of-fit (R2X = 0.664 and R2Y = 0.882), predictability (Q2 = 0.731), and *p*-value (3.12 × 10^−7^), and the cross-validation determined by the permutation test (n = 200) of the PLS-DA model indicated that the model was statistically acceptable. The sample groups were clearly separated according to the fruit development stage. In particular, the Aug. 1 and Nov. 16 samples were clearly separated from each other by t(1), and both samples were discriminated from other samples by t(1) or t(2). Other samples were also clearly separated from each other.

To identify the metabolites contributing to the discrimination among *C. unshiu* samples in the PLS-DA score plot, the variable importance in the projection (VIP) and *p*-values of metabolites were analyzed based on normalized chromatogram intensities (Table 1, Appendix A). A total of 76 metabolites with VIP ≥ 0.72 and *p*-value ≤0.05 were identified. Among them, sugars (fructose, glucose, mannose, sucrose, and inositol), acidic compounds (citric acid, phosphoric acid, malic acid, aconitic acid, threonic acid, quinic acid, and glutaric acid), amino acids and peptides (alanine, serine, proline, threonine, glutamic acid, aspartic acid, glutamine, suberyl glycine, asparagine, arginine, stachydrine, phenylalanine, tryptophan, cyclonatsudamine A, and Gly-Ile-Pro-Tyr-Ile-Ala-Ala), and lipids (phytospingosine, propionic acid, lysophosphatidylethanolamines (LPEs) (C18:3, and C20:0), and lysophosphatidylcholines (LPCs) (C18:3, C16:1, C18:2, C16:0, and C18:0)) were identified as the primary metabolites. In addition, flavonoids (quercetin triglucoside, saponarin, quercetin triglucoside derivatives, kaempferol triglucoside, hesperetin triglucoside, kaempferol-3-rutinoside, isorhamnetin-3-rutinoside, narirutin, hesperidin, didymin, nobiletin, methoxynobiletin, and tangeretin), volatile compounds (α-pinene, β-pinene, α-terpinene, limonene, γ-terpinene, α-terpinolene, linalool, 4-terpineol, α-terpineol, decanal, δ-elemene, α-cubebene, texanol, α-copaene, β-elemene, γ-elemene, α-humulene, germacrene D, α-farnesene, and δ-cadinene), limonoids (zapoterin, xylogranatin K, nomilin, and limonin), and carotenoids (β-carotene and β-cryptoxanthin) were identified as the secondary metabolites. Similar metabolite profiles have also been observed during fruit development in other citrus varieties, including Navel orange [8], Ponkan [10], Persian Lime [11], and Pomelo [24].

### 2.4. Relative Abundance and Metabolic Pathway of Identified Metabolites

To compare the normalized chromatogram intensities of the identified metabolites during fruit development, the changes in metabolite profiles were visualized using heat maps based on the relative abundance of each metabolite (Figure 4), and their fold changes were calculated against those on Aug. 1 (Table 1).

Most sugars, including fructose, glucose, and mannose, accumulate during fruit development. In particular, the relative contents of glucose and fructose on Nov. 16 were approximately 1.7 and 2.0 times, respectively, more than those on Aug. 1. However, there was no significant change in the content of sucrose, the major sugar. A similar result was reported for the later maturation stage of Ponkan [10], in which glucose and fructose increased more rapidly than sucrose during fruit development owing to sucrose degradation by genes related to sucrose metabolism, including invertases, fructokinase, and hexokinase. However, inositol levels decreased with fruit development. Similar results were observed during fruit development in navel oranges [8] and grapefruits [5]. A decrease in inositol during fruit development was also observed in blueberry fruit because of the decreased activity of enzymes, including *Vaccinium corymbosum* (Vc)MIPS2, VcIMP, and VcMIOX [30].

Unlike sugars, all the acidic compounds decreased during fruit development (Table 1 and Figure 4). Among them, the contents of major organic acids, citric acid and malic acid, decreased by approximately 1.6 and 7.4 times, respectively, in the Nov. 16 samples compared to those in the Aug. 1 samples. Additionally, the contents of phosphoric acid, aconitic acid, threonic acid, and quinic acid in the Nov. 16 samples were 3.6, 15.6, 19.3, and 22.3 times, respectively, lower than those in the Aug. 1 samples, whereas glutaric acid was not observed in the Nov. 16 samples. The decrease in acidic compounds during maturation is caused by factors such as reduced ability of the fruit to synthesize acid, dilution as a result of increase in fruit volume and weight, decrease in acid translocation in leaves, and increased respiration [13]. Sugar and acid levels are important factors for citrus maturity and sensory quality [22,31]. In general, the sugar/acid ratio increases during fruit development, and in the case of mandarins, a ratio of 6.5% or higher is set as the maturity index, and a ratio of 14 is considered as the minimum good taste quality [22]. In this study, by comparing the total amount of sugars and acidic compounds in *C. unshiu* fruits during fruit development, the total sugar/total acid ratio was found to increase by 2.6 times during fruit development from Aug. 1 to Nov. 16. Among the amino acids, phenylalanine, suberyl glycine, threonine, alanine, proline, and arginine accumulated during fruit development. In particular, the contents of arginine and proline in the Nov. 16 samples were approximately 40 and 22 times higher than those in the Aug. 1 samples, respectively, whereas the contents of tryptophan, glutamic acid, serine, and stachydrine decreased, and tryptophan and serine were not observed in the Nov. 16 samples. In contrast, asparagine, aspartic acid, and glutamine accumulated on Sep. 14 or Oct. 15 and then decreased. Changes in the amino acid profile of citrus are metabolically related to the tricarboxylic acid cycle, citrate decline, and GABA shunt during fruit development [6,8]. These amino acids are used as nitrogen sources for citrus metabolism and growth, and their profiles are associated with nutritional quality [8].

In addition to the primary metabolites, flavonoids and terpenoids, including limonoids, carotenoids, monoterpenes, and sesquiterpenes, were identified as the main secondary metabolites in *C. unshiu* (Table 1). Flavonoids associated with the plant defense response [32] are synthesized by the conversion of *p*-coumaroyl-CoA into naringenin via chalcone synthase and isomerase [33]. In the present study, all flavonoids decreased during the fruit development of *C. unshiu*. In particular, the contents of major flavonoids including narirutin, hesperidin, nobiletin, and tangeretin in the Nov. 16 samples were approximately 2, 1.3, 2.5, and 2.5 times lower than those in the Aug. 1 samples, respectively. The decrease in citrus flavonoids is the result of an increase in PPO activity [24] and a decrease in gene expression levels associated with chalcone synthases (HCS, CitCHS1, and CitCHS2) during fruit development [34]. A decrease in citrus flavonoids might lead to a decrease in biological activities, including antioxidant, anti-inflammatory, and anti-cancer activities [3,35,36], and may also affect taste quality, such as bitterness and astringency [37,38].

Limonoids, including limonin and nomilin, which are triterpenoids, are associated with citrus bitterness [37,38]. The content of limonin in the Nov. 16 samples was 11 times lower than that in the Aug. 1 samples, while that of nomilin, which increased on Aug. 31, decreased by 8.5 times compared to that in the Aug. 1 samples. These major limonoid changes during fruit development are associated with limonin D-ring lactone hydrolase (LLH) and limonoid glucosyltransferase (LGT) [39]. In Pomelo, the contents of limonin and nomilin were increased by LLH under acidic conditions at the early stage of fruit development and then decreased to glucoside forms by LGT or further biological functions in fruit in the middle of fruit development [39]. By contrast, the contents of minor limonoids, including zapoterin and xylogranatin K, in the Nov. 16 samples were approximately 38 and 157 times higher, respectively, than those in the Aug. 1 sample.

Carotenoids, a group of tetraterpenoids with various biological activities [40], are important citrus pigments expressed in the orange color of the fruit [41]. β-Cryptoxanthin content increased from Aug. 1 (680.93 μg/100 g fresh sample) to Nov. 16 (1989.13 μg/100 g fresh sample), while the β-carotene content gradually increased from Aug. 1 (108.66 μg/100 g fresh sample) to Oct. 15 (251.97 μg/100 g fresh sample) and then increased sharply to six times on Nov. 16 (1566.18 μg/100 g fresh sample). A previous study reported that the β-cryptoxanthin and β-carotene in Satsuma mandarin pulp are accumulated by the lycopene β cyclase and β-carotene hydroxylase [25].

Among the volatile compounds, mono- and sesquiterpenes, known as natural aroma agents with various functional activities, including antioxidant and antimicrobial [42,43], were identified as the main volatile compounds of *C. unshiu*. The content of most monoterpenes decreased during fruit development, except for that of linalool and limonene (Table 1 and Figure 4). In particular, the content of linalool increased by approximately three times during maturation from Aug. 1 to Nov. 16. The content of all sesquiterpenes increased during fruit development stage II (from Aug. 31 to Sep. 14 or Oct. 15) and then decreased. These patterns were different from those of other citrus fruits, such as Persian lime [32] and Navel orange [44]. This metabolic peculiarity may be a defense adaptation of young fruits against herbivores and pathogens during fruit development [45]. In grapefruit plants, monoterpene richness was observed in young flower buds, leaf buds, and enlarged fertilized ovaries, which require protection from herbivores and pathogens [45].

Based on the identified metabolites, a metabolomic pathway associated with fruit development in *C. unshiu* was proposed (Figure 4). It was determined that primary metabolism including sugar, amino acids, and acidic compounds, and secondary metabolism, including flavonoids and terpenoids, mainly changed during fruit development in *C. unshiu*. Similar results have been reported for other citrus varieties, including Navel Orange [8] and Ponkan [10].

### 2.5. Correlation Analysis between Total Phenolic Content, Total Carotenoid, Antioxidant Activity, and Metabolites during Fruit Development

The correlation between antioxidant activity, metabolites, TPC, and TCC at different fruit developmental stages was visualized using a PLS-biplot (Figure 5). The goodness-of-fit (R2X = 0.836; R2Y = 0.930), predictability (Q2 = 0.879), *p*-values (0.625 × 10^−16^), and cross-validation determined by the permutation test (n = 200) of the model indicated that the model was statistically acceptable. The PLS-biplot showed the fruit development of *C. unshiu* and was classified into three groups: Group 1 (Aug. 1), Group 2 (Aug. 31 and Sep. 14), and Group 3 (Oct. 15 and Nov. 16). In Group 1, there was good correlation with TPC, antioxidant activities, flavonoids, monoterpenes, chlorophyll (pheophorbide A), acidic compounds, and some amino acids (serine and tryptophan). In Group 2 there was a good correlation with sesquiterpenes, sucrose, limonin, and some amino acids (glutamine and aspartic acid). In Group 3, glucose, fructose, carotenoids, some amino acids (phenylalanine, threonine, and proline), limonene, and linalool were grouped. These results showed a rich variation in the metabolite profile expressed in citrus, including antioxidant activity, color, and taste at different stages of *C. unshiu* fruit development [6,8,10,24,33].

## 3. Materials and Methods

### 3.1. Sample Preparation

*C. unshiu* fruits were harvested on five different dates (Aug. 1, Aug. 31, Sep. 14, Oct. 15, and Nov. 16, 2018) from *C. unshiu* trees grown in the fields of the Citrus Research Institute (Jeju, Korea) depending on the degree of fruit development (Figure 1). Twenty fruits were randomly selected from five *C. unshiu* trees. After removing the peel by hand, a portion of the flesh of the fruit was freeze-dried, and a portion was crushed. Both freeze-dried and crushed samples were stored at −80 °C until analysis.

### 3.2. Color Measurement

The appearance color values (*L**, *a**, and *b**) of *C. unshiu* during fruit development were measured using a colorimeter (CR-400, Konica Minolta, Inc., Tokyo, Japan), and CCI was calculated using the following equation [21]:CCI = (1000 × *a**)/(*b** × *L**)

### 3.3. Carotenoid Analysis

For carotenoid extraction, fresh sample was crushed using a blender (Hanil Electric Co., Ltd., Bucheon-si, Gyeonggi-do, Korea), and 1 g of fresh sample was mixed with 60 mL of methanol for 1 h. After removing the supernatant, 700 mL of a mixed solvent (hexane:acetone = 1:1) was added to the remaining residue to extract the pigment. The extraction solvent was removed using a rotary vacuum evaporator (Büchi Labortechnik AG, Flawil, Switzerland). After concentration, the dried residue was dissolved in 30 mL hexane and washed three times with distilled water. The residual moisture was removed by adding anhydrous sodium sulfate. TCC was determined using a spectrophotometer (Analytik Jena, Jena, Germany) at 448 nm. β-Carotene (Sigma Chemical Co., St. Louis, Mo, USA) was used as the standard, and TCC was expressed as mg βCE/g fresh sample [46]. The individual carotenoids, including β-cryptoxanthin and β-carotene, were analyzed using high-performance liquid chromatography (Waters Co., Ltd., Milford, MA, USA) equipped with a UV/Visible detector (Waters). A C30 column (YMC Carotenoid S-3, 4.6 mm × 250 mm, 5 μm; YMC Co. Ltd., Kyoto, Japan) was used to separate carotenoids, and the mobile phase was ethyl acetate: acetonitrile (6:4, *v*/*v*) containing 0.2% formic acid at flow rate of 1.0 mL/min with isocratic elution. All analyses were performed in triplicate [47].

### 3.4. Total Phenolic Content and Antioxidant Activity

To measure the TPC and antioxidant activity in *C. unshiu* flesh, 100 g of freeze-dried flesh samples was homogenized with 3 L of 70% aqueous ethanol at room temperature for 120 min under sonication.

TPC was determined using the Folin-Denis method with some modifications [48]. The diluted sample extract (100 µL) was mixed briefly with 10 µL of Folin-Ciocalteu phenol reagent (Sigma Chemical Co.), and 90 µL of 1.5% Na_2_CO_3_ was added. The sample mixture was incubated at 25 °C for 60 min and then its absorbance was measured at 735 nm using a SpectraMax^®^ M3 Multi-Mode Microplate Reader (Molecular Devices, Sunnyvale, CA, USA). Gallic acid was used as a standard and TPC was expressed as mg GAE/g dry extract.

The DPPH free radical scavenging activity was determined as follows [49]: 20 µL of sample was mixed with 180 µL of 0.1 mM DPPH solution (Sigma Chemical Co.). The reaction mixture was incubated at 25 °C for 10 min and the absorbance was measured at 517 nm. The ABTS free radical scavenging activity was determined as follows [50]: ABTS free radical solution was prepared by mixing 7.4 mM ABTS (Sigma Chemical Co.) and 2.6 mM potassium persulfate (1:1, *v/v*) for 15 h at room temperature in the dark. The sample extract (20 µL) was mixed with 180 μL of the ABTS solution, and after incubation at 25 °C for 15 min, absorbance was measured at 734 nm. The scavenging activities of DPPH and ABTS radicals were expressed as IC_50_ values (μg/mL).

### 3.5. Analysis of Metabolites Using GC-MS

GC-MS analysis was carried out according to our previous method with some modifications [51]. To extract non-volatile metabolites, 0.02 g lyophilized samples was dissolved in 70 μL of methoxyamine hydrochloride in pyridine (20 mg/mL) and incubated at 37 °C for 90 min. Afterwards, 70 μL of *N,O*-bis-(trimethylsilyl)-trifluoroacetamide with 1% trimethylchlorosilane (Sigma Chemical Co.) was added for the derivatization of metabolites and reacted at 70 °C for 30 min. The derivatized samples were analyzed using Agilent Technologies 7890A GC System (Agilent Technologies, Santa Clara, CA, USA) equipped with a DB-5MS capillary column (30 m × 0.25 mm, 0.25 μm, Agilent Technologies). The derivatized samples (1 μL) were injected at a split ratio of 1:50. Helium was used as a carrier gas at a flow rate of 1 mL/min, and the oven temperature program for the separation of nonvolatile metabolites was maintained at 70 °C for 2 min, then increased at a rate of 5 °C/min to 150 °C, 3 °C/min to 210 °C, 8 °C/min to 320 °C, and held for 8 min.

Volatile metabolites were analyzed by solid-phase microextraction (SPME)-GC-MS [51]. Fresh sample (0.1 g) was placed in a sealed vial with a septum cap and mixed with 0.5 mL distilled water. The sealed vial was heated at 65 °C for 10 min with stirring at 300 rpm. The volatile metabolites diffused in the headspace were absorbed using a 2 cm length SPME fiber (50/30 μm, DVB/CAR/PDMS Stableflex, Supleco Inc., Bellefonte, PA, USA) at 65 °C for 15 min. The absorbed volatile metabolites were injected to GC (Agilent Technologies) equipped with a DB-WAX column (30 m × 0.25 mm, 0.25 μm, Agilent Technologies). The oven temperature program was maintained at 40 °C for 2 min and then increased at a rate of 5 °C/min to 90 °C, 19 °C/min to 230 °C, and held for 5 min.

The eluent non-volatile and volatile metabolites were detected using a 5975C Inactive XL MSD (triaxial detector) system (Agilent Technologies) with electron ionization at 70 eV. The ion source and interface temperature were 230 °C and 280 °C, respectively. The data was obtained in the full-scan mode with a mass range of 45–550 m/z. Quality control (QC) samples were prepared by mixing all samples and were analyzed once for each sample set.

### 3.6. Analysis of Metabolites Using UPLC-Q-TOF MS

UPLC-Q-TOF MS analysis was carried out according to our previous method with some modifications [51]. To extract metabolites, 0.02 g lyophilized samples was homogenized with 80% aqueous methanol containing terfenadine as an internal standard (IS) using a bullet blender (Next Advence, Troy, NY, USA). After centrifugation, the supernatant was analyzed using a UPLC-Q-TOF MS (Waters) equipped with an Acquity BEH C18 column (2.1 × 100 mm, 1.7 μm, Waters). The column was equilibrated with mobile phase A (0.1% formic acid in water). Metabolites were eluted using a linear gradient with mobile phase B (acetonitrile containing 0.1% formic acid). The column temperature was set at 40 °C, and the eluted metabolites were ionized by positive electrospray ionization (ESI) and detected using Q-TOF MS. Leucine-enkephalin ([M + H] = 556.2771) was used as lock mass. The capillary and sampling cone voltages were set at 3 kV and 40 V, respectively. The desolvation and source temperature were set at 400 and 120 °C, respectively. The MS data were collected with a scan range of 50–1500 m/z, and the MS/MS data were collected using collision energy ramps of 10–40 eV.

### 3.7. Data Processing

The chromatographic intensities obtained by GC-MS were aligned based on retention time, normalized to the IS, and identified using mass spectral libraries (Wiley 9 and NIST 11). The MS data analyzed by UPLC-Q-TOF MS were collected, aligned, and normalized using the MarkerLynx software (Waters). The metabolites were tentatively identified using the online database connected to the UNIFI software (Waters) and the METLIN database (www.metlin.scripps.edu (accessed on 8 January 2022)).

### 3.8. Statistical Analysis

Multivariate statistical analysis of processed MS datasets was performed using SIMCA-P+ v.14.0.1 (Umetrics, Umea, Sweden) and visualized using partial least squares discriminant analysis (PLS-DA). Statistical differences between the obtained data, including TPC, TCC, antioxidant activities, and relative abundances of metabolite variables, were analyzed by one-way analysis of variance (ANOVA) with Duncan’s test using SPSS software (v.25.0, SPSS Inc., Chicago, IL, USA), and their correlation was analyzed and visualized using SIMCA-P+ v.14.0.1.

## 4. Conclusions

In this study, significant changes in TPC, TCC, antioxidant activities, and metabolite profiles of *C. unshiu* were observed during fruit development, and their correlation was analyzed. Additionally, the citrus metabolomic pathway related to fruit development has been proposed. Although citrus peel was not analyzed, and changes in the sensory and functional qualities during fruit development were not investigated in this study, these results indicated that metabolites associated with sensory quality various health benefits were changed during fruit development. In particular, these results will help to understand the changes in chemical profile during citrus fruit development and can provide valuable information for various applications of *C. unshiu* fruits in the citrus industry.

## Figures and Tables

**Figure 1 plants-11-00967-f001:**
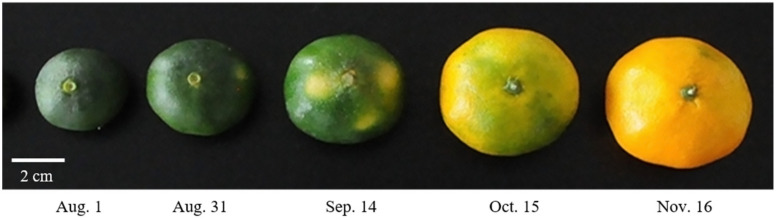
Appearance of *Citrus unshiu* fruit during fruit development.

**Figure 2 plants-11-00967-f002:**
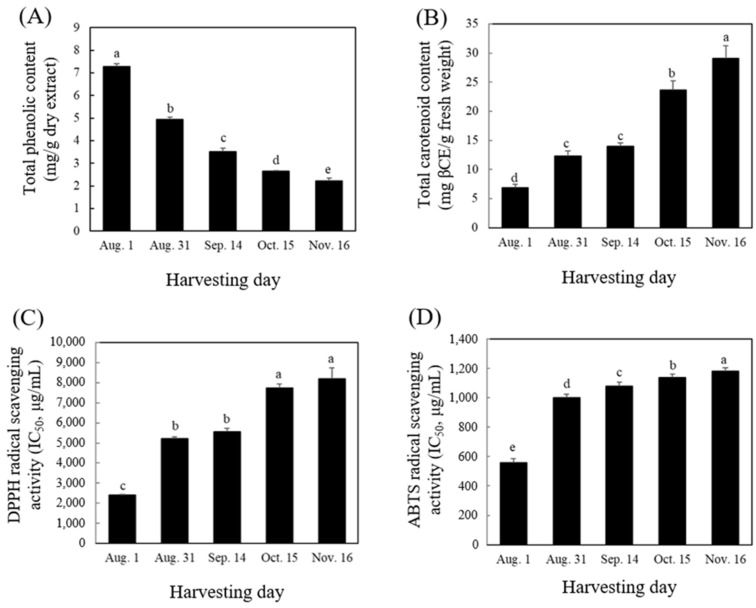
Total phenolic content (**A**), total carotenoid content (**B**), and antioxidant activities (DPPH (**C**) and ABTS (**D**)) of *Citrus unshiu* flesh during fruit development. The scavenging activities of DPPH and ABTS radicals were expressed as half maximum inhibitory concentration (IC_50_). The different letters in each bar indicate significant difference by Duncan’s test at *p* < 0.05.

**Figure 3 plants-11-00967-f003:**
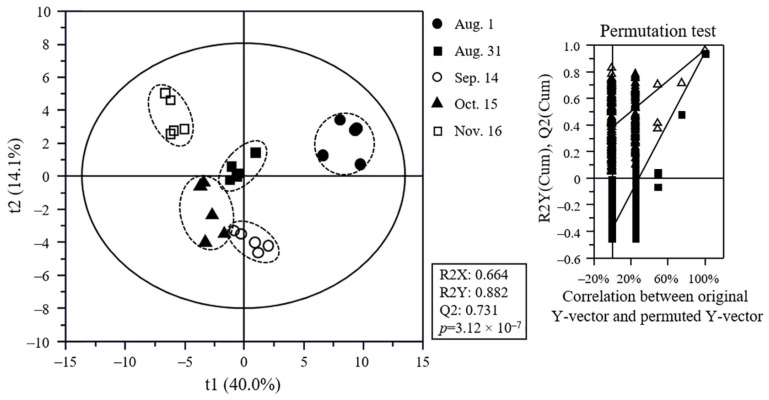
Partial least squares discriminant analysis (PLS-DA) score plot of citrus flesh metabolites and its quality parameters. Metabolites were analyzed using GC-MS, UPLC-Q-TOF MS, and HPLC. The statistical acceptability of the PLS-DA model was evaluated by R2X, R2Y, Q2, and *p*-value and validated by cross validation with a permutation test (n = 200).

**Figure 4 plants-11-00967-f004:**
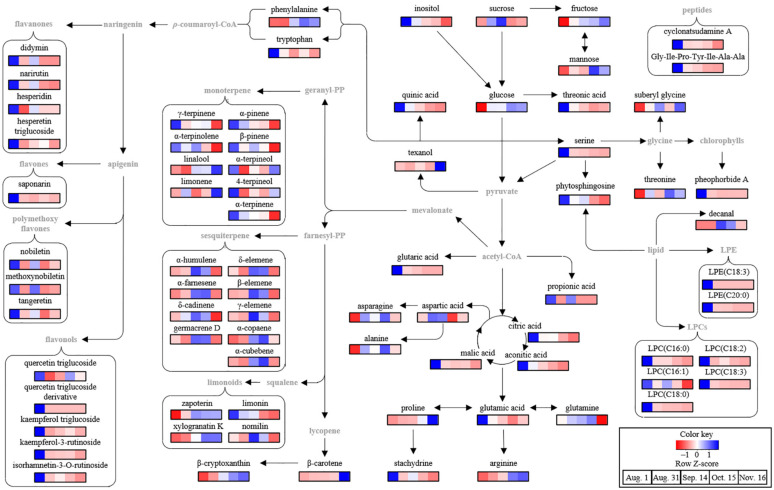
Schematic diagram of the metabolomic pathway associated with the fruit development of *Citrus unshiu*. The metabolomic pathway was retrieved from the KEGG database (https://www.kegg.jp/ (accessed on 17 January 2022)) and drawn with some modifications. The heat maps drawn by R software with ggplot2 showed the relative abundance of identified metabolites. The heat map colors represent the z-score transformed raw data of metabolites with the significant differences among groups. Blue and red indicate the increase and decrease in metabolite levels, respectively.

**Figure 5 plants-11-00967-f005:**
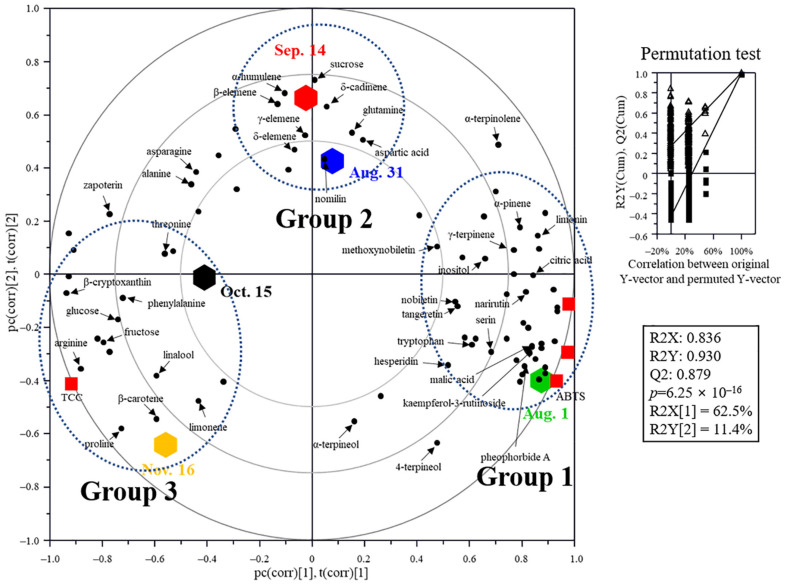
PLS-DA biplot of *Citrus unshiu* flesh during fruit development based on the total phenolic content (TPC), total carotenoid content (TCC), antioxidant activities (DPPH and ABTS), and metabolite profiles. The statistical acceptability was evaluated by quality parameters (R2X, R2Y, Q2, and *p*-value) and validated by cross validation with a permutation test (n = 200).

**Table 1 plants-11-00967-t001:** Identification of the major metabolites from *Citrus unshiu* flesh contributing to the difference of samples in the PLS-DA score plot and their fold change.

	Compounds	VIP	*p*-Value	Fold Change (vs. Aug. 1)			
				Aug. 1	Sep. 14	Oct. 15	Nov. 16
Sugars	fructose	0.93	1.46 × 10^−8^	1.65	1.83	1.88	2.08
	glucose	0.83	1.11 × 10^−9^	1.44	1.43	1.60	1.57
	mannose	1.29	2.33 × 10^−5^	1.15	1.11	1.40	1.28
	sucrose	1.64	2.41 × 10^−4^	1.17	1.26	−1.01	1.01
	inositol	1.59	2.02 × 10^−5^	−1.35	−1.50	−1.56	−1.88
Acidic compounds	citric acid	0.77	2.33 × 10^−5^	−1.33	−1.32	−1.48	−1.61
	phosphoric acid	1.10	9.72 × 10^−5^	−1.57	−2.26	−1.90	−3.55
	malic acid	0.88	8.36 × 10^−7^	−3.54	−4.05	−3.88	−7.40
	aconitic acid	0.89	3.24 × 10^−10^	−2.44	−3.89	−6.44	−15.60
	threonic acid	0.85	1.45 × 10^−7^	−4.11	−8.25	-	−19.26
	quinic acid	0.86	4.40 × 10^−9^	−2.52	−4.25	−10.20	−22.27
	glutaric acid	0.86	1.08 × 10^−7^	−9.08	−17.33	-	-
Amino acids and peptides	arginine	1.27	1.10 × 10^−12^	7.42	14.34	36.42	39.95
	proline	1.59	4.90 × 10^−13^	2.67	3.14	8.46	22.50
	asparagine	1.27	1.60 × 10^−2^	21.92	15.87	25.80	11.52
	threonine	0.84	4.10 × 10^−2^	1.92	1.53	2.30	2.01
	alanine	1.11	2.00 × 10^−2^	2.15	1.81	2.49	1.70
	phenylalanine	1.30	3.47 × 10^−6^	−1.01	1.42	1.62	1.54
	suberyl glycine	1.32	7.26 × 10^−3^	+	++	+	++
	glutamine	1.42	2.10 × 10^−2^	1.13	1.18	1.28	−1.95
	stachydrine	0.86	1.25 × 10^−11^	−1.67	−1.42	−1.75	−2.00
	glutamic acid	0.91	3.59 × 10^−7^	−2.12	−2.63	−4.33	−2.61
	cyclonatsudamine A	0.83	2.57 × 10^−5^	−2.20	−2.14	−2.28	−3.18
	aspartic acid	1.62	8.98 × 10^−4^	−1.07	1.00	−1.25	−3.94
	Gly-Ile-Pro-Tyr-Ile-Ala-Ala	0.84	1.19 × 10^−7^	−3.23	−3.81	−5.44	−6.17
	tryptophan	0.90	6.18 × 10^−3^	−4.15	-	−4.59	-
	serine	0.72	2.43 × 10^−3^	−10.13	−14.03	−37.22	-
Lipids	phytospingosine	0.86	3.15 × 10^−6^	−1.74	−1.57	−2.85	−3.89
	LPC(C16:0)	0.85	3.79 × 10^−12^	−3.02	−3.00	−4.03	−5.43
	LPC(C18:2)	0.74	2.43 × 10^−4^	−6.11	−3.75	−6.16	−8.09
	LPC(C16:1)	0.94	3.22 × 10^−3^	−1.95	−1.28	−2.19	−9.37
	LPC(C18:3)	0.93	2.59 × 10^−13^	−8.36	−12.98	−16.52	−18.18
	LPC(C18:0)	0.82	2.72 × 10^−5^	−11.22	−23.93	−30.64	-
	LPE(C20:0)	0.92	2.95 × 10^−15^	−17.16	−21.47	-	-
	propionic acid	1.06	5.44 × 10^−3^	-	−1.27	-	-
	LPE(C18:3)	0.97	8.14 × 10^−15^	-	-	-	-
Flavonoids	hesperidin	1.12	1.50 × 10^−3^	−1.60	−1.22	−1.34	−1.33
	quercetin triglucoside	1.20	9.02 × 10^−3^	−3.67	−2.35	−1.22	−1.72
	narirutin	0.87	3.67 × 10^−5^	−1.65	−1.38	−1.91	−2.00
	didymin	0.93	3.26 × 10^−4^	−1.95	−1.40	−2.20	−2.23
	methoxynobiletin	1.16	1.40 × 10^−2^	−2.46	−1.04	−2.66	−2.34
	tangeretin	0.81	3.70 × 10^−2^	−2.45	−1.72	−4.43	−2.49
	nobiletin	1.03	8.82 × 10^−3^	−3.64	−1.45	−4.54	−2.53
	kaempferol triglucoside	0.81	8.76 × 10^−3^	−3.01	−2.56	−2.24	−2.86
	saponarin	0.96	5.48 × 10^−8^	−3.00	−3.31	−2.75	−3.16
	isorhamnetin-3-rutinoside	0.81	3.63 × 10^−6^	−3.28	−2.53	−3.54	−5.28
	kaempferol-3-rutinoside	0.90	5.58 × 10^−8^	−5.79	−5.16	−4.96	−10.27
	hesperetin triglucoside	0.87	2.43 × 10^−3^	-	−5.58	−3.64	-
	quercetin triglucoside derivative	0.91	5.86 × 10^−8^	-	-	-	-
Monoterpenes	linalool	1.58	6.37 × 10^−5^	−1.68	1.97	1.94	3.06
	limonene	1.52	9.89 × 10^−6^	1.08	−1.03	1.01	1.16
	α-terpineol	1.42	8.64 × 10^−3^	−5.75	−1.65	−2.17	−1.04
	4-terpineol	1.43	9.65 × 10^−6^	−6.79	−2.29	−3.21	−1.57
	α-terpinolene	1.17	1.20 × 10^−5^	−1.23	−1.05	−1.44	−2.25
	γ-terpinene	1.12	1.31 × 10^−5^	−1.81	−1.60	−1.56	−3.69
	β-pinene	1.08	1.48 × 10^−4^	−1.77	−1.32	−1.73	−10.63
	α-pinene	1.14	4.33 × 10^−5^	−1.51	−2.27	−2.54	-
	α-terpinene	1.33	2.84 × 10^−11^	−1.61	−1.91	−2.07	-
Sesquiterpenes	α-copaene	1.13	1.40 × 10^−2^	1.79	6.23	7.60	3.30
	α-farnesene	1.56	2.21 × 10^−5^	2.12	6.70	6.97	−1.25
	α-cubebene	1.50	7.27 × 10^−4^	−2.12	3.86	5.07	−1.62
	β-elemene	1.71	4.41 × 10^−8^	1.08	3.10	2.60	−2.27
	α-humulene	1.65	1.48 × 10^−6^	1.20	3.16	2.48	−2.41
	δ-elemene	1.69	1.62 × 10^−6^	−2.55	2.34	2.22	−3.30
	germacrene D	1.48	1.16 × 10^−3^	−4.95	3.11	3.01	-
	γ-elemene	1.43	9.92 × 10^−4^	−1.45	4.28	2.42	-
	δ-cadinene	1.35	7.93 × 10^−3^	1.62	2.50	2.06	-
Limonoids (triterpenes)	xylogranatin K	1.20	3.56 × 10^−7^	17.86	86.26	157.40	157.01
	zapoterin	1.04	2.32 × 10^−5^	23.71	41.43	39.22	38.73
	nomilin	1.26	2.30 × 10^−2^	−1.84	2.76	1.65	−8.51
	limonin	0.92	5.33 × 10^−6^	−1.68	−1.84	−5.99	−11.04
Carotenoids (tetraterpenes)	β-carotene	1.76	7.46 × 10^−16^	1.75	1.91	2.32	14.41
	β-cryptoxanthin	1.26	2.49 × 10^−10^	1.48	2.22	2.68	2.92
Other	pheophorbide A	0.90	5.77 × 10^−8^	−15.16	−36.10	−34.78	−28.29
	texanol	1.37	3.80 × 10^−2^	-	1.93	-	14.75
	decanal	1.50	9.80 × 10^−5^	−1.03	7.13	5.70	2.31

VIP, variable importance in the projection; + and ++, newly generated and significantly higher than + (*p* < 0.05), respectively; -, not detected. *p*-values were analyzed by Duncan’s test.

## Data Availability

Data are contained within the article and Appendix A.

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
