# Peer review of "Metabolomic Profiling of Citrus unshiu during Different Stages of Fruit Development"

_plants, 2022, doi:10.3390/plants11070967_

Round 1
Reviewer 1 Report
The author carried out the related research on metabolites in different development stages of citrus fruit. There are the following problems: 1 What is the significance of this study? For the citrus industry? Or for food related industries? Or what scientific problems can be revealed? 2. How to choose the starting point of development stage, what is the basis, and why not start after flowering? 3. The research material of this paper is mainly the pulp of the fruit. However, the content of phenolic compounds and flavonoids in the pulp of citrus is very low, which seems to be meaningless?
Author Response
Comment 1: What is the significance of this study? For the citrus industry? Or for food related industries? Or what scientific problems can be revealed?
Response: Thank you for your comment. To clarify, the following sentence has been added to Abstract and Introduction.
- Abstract: This can provide vital information for various applications in the unshiu industry.
- Introduction: Therefore, in this study, flesh primary and secondary metabolites of unshiu fruits harvested at five different developmental stages were analyzed using ultra-performance liquid chromatography-quadrupole time-of-flight mass spectrometry (UPLC-Q-TOF MS), gas chromatography-mass spectrometry (GC-MS), and high-performance liquid chromatography (HPLC) to provide chemical information for various applications in the C. unshiu industry.
Comment 2: " How to choose the starting point of development stage, what is the basis, and why not start after flowering?
Response: C. unshiu is mostly consumed as a mature fruit, but the immature fruit is also consumed in various products, such as vinegar, jelly, jams, and yogurt. Therefore, considering the size for general commercial use, the starting point was set to Aug. 1. To clearly explain the commercial products of immature C. unshiu, the fallowing sentence has been added to Introduction section.
‘In particular, C. unshiu, which is bud burst in middle of Mar., blooms in May, and is harvested after middle of Nov. in East Asia [18], is mostly consumed as a mature fruit, but the immature fruit is also consumed in various products, such as vinegar, jelly, and jams [19,20].’
Comment 3: The research material of this paper is mainly the pulp of the fruit. However, the content of phenolic compounds and flavonoids in the pulp of citrus is very low, which seems to be meaningless?
Response: As the reviewer commented, the flavonoid content of the flesh is low. However, in addition to flavonoids, the pulp contains carotenoids and limonoids, which we consider significant.
Reviewer 2 Report
In this study, the authors analyze the TPC, TCC profiles, antioxidant activities and metabolites of the flesh of C. unshiu during fruit development and deduce correlations. the authors' results provide a better understanding of the changes in the chemical profile during the development of citrus fruits and thus provide information for a better harvest period.
The introduction must however be improved in order to better position the study in relation to work already published by other authors. The results are consistent and perfectly well presented and above all well discussed in relation to current knowledge.
As indicated by the authors, analyzes of citrus peel and sensory & functional analyzes are missing from the study. But the many analyzes carried out compensate for this lack.
I propose a publication of this study after a thorough proofreading in order to correct some minor errors, particularly typographical.
Author Response
Comment 1: The introduction must however be improved in order to better position the study in relation to work already published by other authors.
Response: Thank you for your comment. We have improved the introduction by citing published work by other authors as follows (lines 63-67).
‘Although previous studies of C. unshiu have shown the changes in the phytochemical profiles of C. unshiu according to the growing regions [14] and the post-harvest process [15], and comparison of the profiles with other citrus varieties [16], the changes in the phytochemical profile of C. unshiu at different stages of fruit development stages have not yet been analyzed.’
Comment 2: I propose a publication of this study after a thorough proofreading in order to correct some minor errors, particularly typographical.
Response: We corrected the minor errors presented as follows.
Line 35: climnical -> clinical
Lines 135, 147, 348, 349, 362, 394: GC/MS -> GC-MS
Line 135: UPLC-QTOF -> UPLC-Q-TOF
Line 303: fruit -> fruits
In addition, the reference form has been modified as prescribed:
No. 2, 4, 8, 10, 11, 22, 28, 34‒38, 42, 46, 47
Reviewer 3 Report
Introduction:
To improve the contextualization, I would recommend the authors complement the introduction part with the phenology of C.unshiu (flowering period, maturation time, and harvest collection time).
Results and Discussion:
Why are the morphological parameters of the fruits not considered (weight, color, diameter, etc.)? the authors only refer to the bibliography, but they are parameters that affect it... (Ln 72-80).
Ln 110 - 113. Complement the discussion of the results with a direct comparison with the bibliographical references, and specify the maturity stage
Ln 131 - 150. Are there references that support the results or that can be compared/discussed/explained?
Materials and methods
Ln 276 - 281. Specify the growing/cultivation conditions, the variety, rootstock (if any), the fruit collection, and peeling methodology. Since in later sections a fresh sample is used (I guess juice), specify the process of obtaining it.
Ln 283 and Ln 334. Specify how the sample fresh is obtained.
Ln 285. specify the experimental conditions for obtaining the "dry residue"
Ln 372 - 378. Why the authors have the ANOVA not been performed? and Principal Component Analysis (PCA)?
Conclusions.
Improve and complete the conclusions with specific data of the results, since the conclusions presented are a general summary of the work and are presented repeatedly (verbatim) with the abstract of the article
Author Response
Comment 1: To improve the contextualization, I would recommend the authors complement the introduction part with the phenology of C. unshiu (flowering period, maturation time, and harvest collection time).
Response: We mentioned the phenology of C. unshiu from bud burst to harvest in the introduction as follows (lines 60-63):
‘In particular, C. unshiu, which is bud burst in middle of Mar., blooms in May, and is harvested after middle of Nov. in East Asia [18], is mostly consumed as a mature fruit, but the immature fruit is also consumed in various products, such as vinegar, jelly, and jams [19,20].’
Comment 2: Why are the morphological parameters of the fruits not considered (weight, color, diameter, etc.)? the authors only refer to the bibliography, but they are parameters that affect it... (Ln 72-80).
Response: Thank you for your comment. We added morphological parameters (weight, size, and color) to the supplementary data (Table S1), and mentioned them in the manuscript as follows (linse 82-90):
Weight, size, and color of C. unshiu fruit were analyzed according to fruit development (Figure 1 and Table S1). The fruit weight of C. unshiu increased from 31.37 ± 3.79 g (Aug. 1) to 111.45 ± 11.52 g (Nov. 16), while the width and length were increased from 3.47 ± 1.92 and 4.16 ± 1.68 cm (Aug. 1) to 6.44 ± 2.71 and 5.20 ± 2.24 cm (Nov. 16), respectively. The color of C. unshiu changed dark green (Aug. 1) to partially orange (Oct. 15) to full orange (Nov. 16). In addition, the citrus color index (CCI), a standard parameter used to determine the color of citrus [20], showed negative values (‒15.03 to –16.66) from Aug. 1 (‒15.03) to Sep. 14 (–16.66), whereas it showed positive values on Oct. 15 (0.88) and on Nov. 16 (11.59).
In addition, color measurement method (section 3.2) has been added as follows (lines 304-308)
3.2. Color measurement
The appearance color values (L*, a*, and b*) of C. unshiu during fruit development were measured using a colorimeter (CR-400, Konica Minolta, Inc., Tokyo, Japan), and CCI was calculated using the following equation [20]:
CCI = (1,000 × a*)/(b* × L*)
Comment 3: Ln 110 - 113. Complement the discussion of the results with a direct comparison with the bibliographical references, and specify the maturity stage
Response: The change in antioxidant activities according to maturity stages has been discussed in comparison with previously reported data. To clarify, the fallowing sentences have been modified. (lines 129-133)
‘The decrease in antioxidant activity during fruit development was positively correlated with a decrease in phenolic compounds. Similar results have been observed during the fruit development stages in various citrus fruits, including Pomelo [22], sweet orange [25], Chinese native varieties [26], and Navel orange [27]. The decrease in phenolic compounds according to the fruit development and maturity stage led to a decrease in their antioxidant activities.’
Comment 4: Ln 131 - 150. Are there references that support the results or that can be compared/discussed/explained?
Response: We have added four references to compare with the results (lines 170-172).
‘Similar metabolite profiles have been observed during fruit development in other citrus varieties, including Navel orange [11], Ponkan [13], Persian Lime [14], and Pomelo [22].’
Comment 5: Ln 276 - 281. Specify the growing/cultivation conditions, the variety, rootstock (if any), the fruit collection, and peeling methodology. Since in later sections a fresh sample is used (I guess juice), specify the process of obtaining it.
Response: Thank you for your comment. Variety and the fruit collection method have already been mentioned in lines 298, and 300-301. We have collected fruits from C. unshiu trees grown in the fields of the Citrus Research Institute and their peels were removed by hand. To clearly explain the method, the ‘sample preparation’ section has been modified as follows.
‘C. unshiu fruits were harvested on five different dates (Aug. 1, Aug. 31, Sep. 14, Oct. 15, and Nov. 16, 2018) from C. unshiu trees grown in the fields of the Citrus Research Institute (Jeju, Korea) depending on the degree of fruit development (Figure 1). Twenty fruits were randomly selected from five C. unshiu trees. After removing the peel by hand, some of the flesh was freeze-dried, and the rest was crushed. Both the freeze-dried and the crushed samples were stored at ‒80 °C until analysis.’
Comment 6: Ln 283 and Ln 334. Specify how the sample fresh is obtained.
Response: The fresh sample was obtained by crushing the C. unshiu flesh. The method has been mentioned in lines 310-311.
‘For carotenoid extraction, fresh sample was crushed using a blender (Hanil Electric Co., Ltd., Gyeonggi-do, Korea), and 1 g of fresh sample was mixed with 60 mL of methanol for 1 h.’
Comment 7: Ln 285. specify the experimental conditions for obtaining the "dry residue"
Response: The dry residue was obtained using a rotary vacuum evaporator. The method has been mentioned on lines 313-314.
‘The extraction solvent was removed using a rotary vacuum evaporator (Büchi Labortechnik AG, Flawil, Switzerland).’
Comment 8: Ln 372 - 378. Why the authors have the ANOVA not been performed? and Principal Component Analysis (PCA)?
Response: We performed one-way analysis of variance (ANOVA). “(ANOVA)” has been mentioned (line 405). For multivariate statistical analysis of metabolites, we performed PCA and PLS-DA. PCA has been used to the Hotelling’s T2-test before data visualization using PLS-DA. However, we only mentioned the data visualization method using PLS-DA in the manuscript because there was no samples located outside of 95% confidence ellipse analyzed by the Hotelling’s T2-test using PCA.
Comment 9: Improve and complete the conclusions with specific data of the results, since the conclusions presented are a general summary of the work and are presented repeatedly (verbatim) with the abstract of the article
Response: Thank you for your comment. We have modified the conclusion section as follows.
In this study, significant changes in TPC, TCC, antioxidant activities, and metabolite profiles of C. unshiu were observed during fruit development, and their correlation was analyzed. Additionally, the citrus metabolomic pathway related to fruit development has been proposed. Although citrus peel was not analyzed, and changes in the sensory and functional qualities during fruit development were not investigated in this study, these results indicated that metabolites associated with sensory quality various health benefits were changed during fruit development. In particular, these results will help to understand the changes in chemical profile during citrus fruit development and can provide valuable information for various applications of C. unshiu fruits in the citrus industry.
Round 2
Reviewer 1 Report
This article has carried out in-depth research on the changes of metabolites in the developmental stages of Citrus unshiu. Multiple types of metabolites are detected and the analysis is comprehensive, which makes this article valuable for reference. However I think this article still needs some improvement before publication, and my detailed comments are as follows:
- In the introduction, the role of the metabolites should be clearly introduced. You have mentioned that some metabolites such as phenolics, flavonoids, limonoids, carotenoids and volatile terpenes are related with health. But in addition to these biochemicals , many of the metabolites you covered in this study are related to the color and taste of the fruit, including sugars, acids, amino acids, and carotenoids, which can determine the edible value and commodity value of Citrus unshiu. I think the significance and research value of these metabolites should be emphasized in the introduction.
- I think the innovation of the article should be highlighted in the introduction. For example ,why choose Citrus unshiu as materials, is there something special? Why focus on metabolite changes during developmental stages? Why pay attention to the antioxidant activity of the fruit pulp? I hope that you can show the innovation of your study.
- I think your article lacks some concluding discussions. The part of discussions just compares the similarities and dissimilarities between the results obtained in this study and previous studies, making the conclusions a little monotonous. I hope that you can put forward some helpful tips on research and production according to your study, such as, at what stage the fruit color and taste is the best to appear on the market, at what stage the fruit has the strongest antioxidant activity so that can be used in the field of medicine, and whether the study of metabolite changes with developmental stage can help to find some key genes. The above are just some of my ideas and I just hope that you can enrich your discussion to make the research more meaningful.
Author Response
Comment 1: In the introduction, the role of the metabolites should be clearly introduced. You have mentioned that some metabolites such as phenolics, flavonoids, limonoids, carotenoids and volatile terpenes are related with health. But in addition to these biochemicals, many of the metabolites you covered in this study are related to the color and taste of the fruit, including sugars, acids, amino acids, and carotenoids, which can determine the edible value and commodity value of Citrus unshiu. I think the significance and research value of these metabolites should be emphasized in the introduction.
Response: Thank you for your comment. We mentioned metabolites related to citrus quality in the introduction section as follows (line 47):
During the fruit development stage of citrus, its qualities, including appearance, nutrition, taste, and biological activity, are significantly changed by changes in metabolite profiles such as sugars, acids, amino acids, flavonoids, and terpenes.
Comment 2: I think the innovation of the article should be highlighted in the introduction. For example, why choose Citrus unshiu as materials, is there something special? Why focus on metabolite changes during developmental stages? Why pay attention to the antioxidant activity of the fruit pulp? I hope that you can show the innovation of your study.
Response: We have improved the introduction section based on your comments as follows:
Lines 45-47: During the fruit development stage of citrus, its qualities, including appearance, nutrition, taste, and biological activity, are significantly changed by changes in metabolite profiles such as sugars, acids, amino acids, flavonoids, and terpenes.
Lines 59-62: In particular, C. unshiu, which is bud burst in middle of Mar., blooms in May, and is harvested after middle of Nov. in East Asia [15], is mostly consumed as a mature fruit, but the immature fruit is also consumed in various products, such as vinegar, jelly, and jams to produce citrus products having strong antioxidant activity [16,17,18].
Comment 3: I think your article lacks some concluding discussions. The part of discussions just compares the similarities and dissimilarities between the results obtained in this study and previous studies, making the conclusions a little monotonous. I hope that you can put forward some helpful tips on research and production according to your study, such as, at what stage the fruit color and taste is the best to appear on the market, at what stage the fruit has the strongest antioxidant activity so that can be used in the field of medicine, and whether the study of metabolite changes with developmental stage can help to find some key genes. The above are just some of my ideas and I just hope that you can enrich your discussion to make the research more meaningful.
Response: Thank you for your comment. Based on your comments, we have improved some lack of the Results and Discussion section as follows:
Line 85: In general, matured mandarin has a CCI value of 5 or higher [22].
Lines 123-124: This result indicated that immature fruits are suitable for the production of citrus products having high antioxidant activity.
Lines 186-189: A similar result was reported for the later maturation stage of Ponkan [10], in which glucose and fructose increased more rapidly than sucrose during fruit development owing to sucrose degradation by genes related to sucrose metabolism, including invertases, fructokinase, and hexokinase.
Lines 250-252: A previous study reported that the β-cryptoxanthin and β-carotene in Satsuma mandarin pulp are accumulated by the lycopene β cyclase and β-carotene hydroxylase [25].
